# Further Inspection: Integrating Housing Code Enforcement and Social Services to Improve Community Health

**DOI:** 10.3390/ijerph182212014

**Published:** 2021-11-16

**Authors:** Katharine Robb, Ashley Marcoux, Jorrit de Jong

**Affiliations:** Ash Center for Democratic Governance and Innovation, Harvard Kennedy School, Cambridge, MA 02138, USA; ashley_marcoux@hks.harvard.edu (A.M.); jorritdejong@hks.harvard.edu (J.d.J.)

**Keywords:** housing, code enforcement, urban health, housing inspection, social services, collaboration

## Abstract

As a result of working inside homes, city housing inspectors witness hidden and serious threats to public health. However, systems to respond to the range of problems they encounter are lacking. In this study, we describe the impact and enabling environment for integrating a novel Social Service Referral Program within the Inspectional Services Department in Chelsea, MA. To evaluate the first eight months of the program, we used a mixed-methods approach combining quantitative data from 15 referrals and qualitative interviews with six key informants (inspectors, a case manager, and city leadership). The most common services provided to residents referred by inspectors were for fuel, food, and rent assistance; healthcare; hoarding; and homelessness prevention. Half of referred residents were not receiving other social services. Inspectors reported increased work efficiency and reduced psychological burden because of the program. Interviewees described how quality of life improved not only for referred residents but also for the surrounding neighborhood. A simple referral process that made inspectors’ jobs easier and a trusted, well-connected service provider funded to carry out the work facilitated the program’s uptake and impact. Housing inspectors’ encounters with residents present a unique opportunity to expand the public health impact of housing code enforcement.

## 1. Introduction

### 1.1. Housing, Health, and the Role of the Housing Inspector

Housing is more than physical shelter; it is a powerful social determinant of health [1]. Poor housing is associated with health outcomes as diverse and far reaching as cardiovascular disease, lead poisoning, mental illness, and infectious disease [2,3,4,5,6]. These risks stem from physical problems, such as insect infestations causing asthma exacerbations [7], financial problems that result in poor property maintenance [8], and social problems, such as overcrowded conditions straining interpersonal relationships [9].

A primary strategy for breaking the link between poor housing and poor health is the enforcement of housing codes [10]. Housing codes stipulate minimum health and safety standards in rental housing. Inspectors enforce these codes by visually inspecting inside homes, identifying violations, and issuing citations to achieve compliance. Housing inspection has transformative potential to improve the health of households and communities [10]. For example, smoke detector installation can dramatically reduce loss of life and livelihood from fire [11], while extermination of insects can reduce asthma-related emergency room visits [12].

Housing inspectors, by nature of their work inside homes, are often the only public officials to witness many serious health risks—some of which constitute code violations, but many of which do not. These risks include inadequate protection from the elements, hoarding disorders, and overcrowded conditions. While inspectors are in a unique position to intervene, they lack systems and training to respond to the range of public health problems they encounter: the only action available to them is enforcement through citations. While some housing-related problems are effectively addressed through enforcement alone (e.g., a landlord not providing smoke detectors), others require service provision paired with enforcement (e.g., when poverty leads to deteriorating, unsafe conditions). Pairing code enforcement (an obligation encounter that imposes duties) with service provision (an encounter where assistance is provided) is an opportunity to address root causes of housing-related problems and improve the effectiveness, efficiency, and equity of code enforcement.

### 1.2. Housing and Multisector Collaboration

There is growing evidence that collaboration between local governments, public health, and social service institutions has a synergistic impact on health outcomes [13]. Collaboration often requires these institutions to reimagine their encounters with one another and the populations they serve. Within housing inspection, collaboration can take the form of compliance assistance (e.g., programs for low-income landlords to remediate lead paint), service provision (e.g., relocation in the event of eviction) or may originate outside of inspection departments. For example, in Boston, MA, through the Breathe Easy at Home program, health professionals make referrals to housing inspectors if they suspect housing conditions are contributing to a child’s asthma. Since 2013, the program has helped thousands of renters work with landlords to address housing-related asthma triggers that violate the housing code [14]. Newark, NJ established a task force of city agencies in 2014 to coordinate responses to code violations when the health of residents was at risk. This not only helped facilitate intergovernmental collaboration to solve other problems but provided more streamlined services to residents [15].

Collaborations like these can make housing code enforcement a more effective means to improve public health through early intervention. In fact, when housing inspection began in the late 1800s, code enforcement and public health went hand-in-hand. In modern times, formal collaboration between housing inspection and health and social service agencies are not the norm and existing programs are limited in the range of problems they address. Inherent tension between service-oriented professions and compliance-oriented professions can stymie collaboration [16]. Yet, given the powerful role housing plays in shaping health, inspectors’ encounters with residents represented an unrealized opportunity to address pressing housing and health problems and provide support to difficult-to-reach populations.

### 1.3. Placing Modern Housing Inspections in Historical Context

In the mid-19th century, rapid population growth among the urban poor and lack of affordable, adequate housing gave rise to overcrowded and severely substandard housing conditions [17]. Conditions like these motivated city boards of health to establish housing codes [18]. The role of the housing inspector, often a medical officer, involved education on home hygiene and health behaviors, as well code enforcement [17,19]. The interaction blended obligation to comply with codes alongside service provision, where needed.

As the 20th century progressed, budgets for public health departments in the U.S. shrank [20]. With less political will for comprehensive housing and health programs and major threats to mortality reduced, health departments turned their focus to single-issue programs, such as lead removal or tuberculosis control [20]. The approach to housing and health became siloed [18]. Once part of health departments, inspection became its own department in many cities [21]. The role of the housing inspector became obligation-focused and detached from public health, making it difficult to coordinate service across sectors [21]. While housing conditions in U.S. cities have dramatically improved over the last 150 years, persistent housing-related public health problems remain. In some low-income communities, renters still experience many of the health and social challenges faced in tenements in the late 1800s (e.g., poor ventilation, overcrowding) [22,23,24,25] in addition to modern risks to public health (e.g., the opioid epidemic, COVID-19 infection) [26,27]. A more dynamic approach to housing inspection that renews and leverages its historical connection with public health is needed.

### 1.4. Objective

The objective of this study is to describe the impact and enabling environment for integrating a novel Social Service Referral Program within the Inspectional Services Department in Chelsea, MA. Impact is measured in the quantity and quality of services provided to residents through the program, as well as how the program changed work processes for housing inspectors and social service providers.

## 2. Materials and Methods

### 2.1. Setting

Chelsea is a small, densely populated, demographically diverse city located just outside Boston. Per-capita income is USD 23,240/year, making Chelsea one of the poorest cities in Massachusetts [28]. The majority of residents are racial and ethnic minorities (78%) and almost half are foreign-born (46%) [28]. Half of the housing stock is two-to-four-family homes. Most homes were built over a century ago [29]. Almost 70% of residents rent, and many face overcrowded and unsafe living conditions [29]. Chelsea is disproportionately impacted by pollution and chronic disease in comparison with the rest of the state [30,31]. At the same time, Chelsea’s small size and high population density foster a close-knit community with a robust network of social service providers and grassroots activists, giving rise to forward-thinking approaches to solving these problems [31].

In 2015, Chelsea City Hall began a proactive rental housing inspection program to improve housing conditions [32]. Inspectors’ jobs, which require a minimum of a high-school education and job-specific certification, are focused on assessing physical structures, not the people living inside [21]. Yet, inspectors encounter social problems on a regular basis, including mental illness, substance use disorders, and eviction [33].

In 2017, the Harvard Kennedy School’s Innovation Field Lab partnered with five Massachusetts cities, including Chelsea, to develop innovative solutions to intractable housing problems [34]. Based on goals expressed by city leadership, namely the need to improve processes for inspectors and social outcomes for residents, the idea for the Social Service Referral Program was born. The authors of this paper were involved in the conceptualization, design, and early change management work to establish the program in collaboration with inspectors and city leadership.

In 2019, the City of Chelsea, MA implemented a novel social service referral program within the housing inspectional services department. When inspectors identify a social or health problem that cannot be addressed through code enforcement alone, they can quickly and easily connect residents with a social service case manager. The case manager connects residents with a range of services from substance use disorder treatment to financial assistance with housing repairs.

Significant change management work occurred prior to the initiation of the program. The Social Service Referral Program required more than collaboration across diverse agencies; it required pairing traditionally very separate types of encounters—one which imposes duties and one which provides assistance [16,21,34]. Building consensus on the causes and consequences of the problem and acknowledging the tension between obligation and service encounters was particularly important, especially in a context where residents are vulnerable and/or have limited trust in government and where a fundamental shift in the perspective of front-line staff (in this case, inspectors) was required [16]. Gaining trust and buy-in from inspectors as agents of, rather than targets of, the change in processes was essential. This process is described in greater detail in previously published work on the Social Service Referral Program [33,35].

Prior to the adoption of the program, a pilot program existed for several months to show proof of concept. In the pilot, inspectors could make referrals to a Community Engagement Specialist within City government who had connections with local social service agencies. The pilot convinced City leadership that inspectors would make referrals and residents would accept them. The pilot softened inspectors’ main objection to the program by reducing, instead of increasing, their day-to-day workload. However, the program lacked funding, accountability, and ultimately sustainability [35]. Relying on good intentions was insufficient to address the complex needs of residents, but the pilot demonstrated the idea was viable.

Chelsea’s Social Service Referral Program launched in earnest in July 2019, when the City contracted a local social service agency, CAPIC (Community Action Programs Inter-City Inc., Chelsea, MA, USA) to provide services to in-need residents identified during housing inspection. CAPIC had an existing contractual relationship with the City to provide crisis and outreach services but was not previously connected with inspectors. The agency serves an average of 14,000 low-income individuals and families annually in Chelsea and nearby cities [36]. Their mission is to eradicate root causes of poverty through a range of programs. Typically, clients of CAPIC visit their offices to apply for assistance or engage in one of their programs (after school care, tax preparation, substance use disorder support). Formal collaboration between the Innovation Field Lab and Chelsea City Hall concluded a month after contract initiation, at which time City Hall led the program independently. The contract has been renewed each year since. Services provided include CAPIC’s in-house programs (e.g., emergency assistance, childcare, weatherization), as well as linkages to other agencies (e.g., mental health programs, eldercare).

The Social Service Referral Program has four steps:During routine inspections, inspectors identify residents in need of support beyond what inspectors can provide, ranging from crisis assistance (e.g., a family being evicted) to vital access to basic needs (e.g., heating fuel). Referrals can be made for landlords as well as tenants. No formal screening process is used in making referrals.To make a referral, inspectors obtain consent from residents and call a designated case manager at the social service agency, CAPIC. Inspectors received training from CAPIC’s case manager on this process.A case manager then contacts the resident, most often meeting them at their home, to determine what type of support is needed. If a resident accepts, the case manager connects them with services and provides follow-up care. Housing inspectors follow up to ensure housing code violations, if present, are corrected.CAPIC shares outcomes and progress with inspectors and the City Manager via monthly updates and quarterly reports.

### 2.2. Data Sources

To evaluate the impact of the first eight months of the referral program (July 2019 through February 2020), we used a mixed methods approach combining quantitative data from the first 15 referrals with qualitative interviews with six key informants.

### 2.3. Quantitative Data

Case managers collected demographic data on referred residents, in addition to data on the number of refusals, services provided, and follow-up activities. We describe these data in summary statistics.

### 2.4. Qualitative Data

Semi-structured qualitative interviews (see Appendix A for interview guides) were conducted with inspectors (*n* = 3 of 5); the lead social service case manager; and Chelsea’s City Manager and Deputy City Manager. Interviewees were selected based on their involvement in the Social Service Referral Program and interviews lasted 30 to 60 min. Interviews were completed in Fall 2020 via phone (*N* = 1), videoconferences (*N* = 4) or in person (*N* = 1) by the first and second authors. Interviewees were asked questions about the perceived impact of the program on their work and on residents, as well as conditions that enabled the program’s adoption. The study protocol originally included interviews with referred residents, with contact between residents and the research team facilitated through CAPIC’s case manager. However, this connection was not possible due to CAPIC’s ongoing response to the COVID-19 crisis.

Interviews were recorded and transcribed. Thematic analysis [37] was used to code transcripts in Dedoose (version 8.0.35) to identify themes related to the impact, challenges, and enabling conditions. Both the first and second authors analyzed all interviews. Sub-themes were identified after initial coding and a codebook was developed. The codes included deductive codes from the topics in the interview guides and inductive codes from the data itself.

## 3. Results

### 3.1. Referral Characteristics

Inspectors referred 15 residents to social services. Half of residents referred were women and 60% were over 60 years-old (Table 1). Forty percent were White (non-Hispanic/Latino); 27% were Hispanic/Latino. Twenty percent had children under five years; 53% were classified by a case manager as physically disabled.

More than half (60%) accepted services upon the first offer; 27% accepted after two or more offers; 7% declined services (Table 2). For 20%, the referral was their first connection to any kind of social service; 33% had been previously connected to services; and 47% were currently receiving other forms of assistance.

Residents received, on average, nine visits or phone calls with a case manager (median 7, range 1–33) and three types of services (range 1–5). Figure 1 shows the type and number of services provided. Most services were for heating fuel and food assistance, followed by help accessing healthcare (including substance use treatment and mental healthcare), hoarding clean-up, homelessness prevention, and rent assistance. 

### 3.2. Qualitative Interview Themes

#### 3.2.1. Challenges Inspectors Faced before the Referral Program

Inspectors reported routine encounters with families or individuals in crisis or in need of support. They described finding families living on enclosed porches, unfinished basements, or closets; the unhygienic conditions of homes where residents hoarded food or animals; bare cupboards; no heat in the winter; and seniors in need of assistance to live safely at home. Inspectors recounted how these problems impeded their work because they limited residents’ ability to resolve violations and caused repeat inspections. This reduced time available for other inspections and many violations were only temporarily resolved. Before the referral program, when inspectors tried to support residents in crisis, it often resulted in lengthy engagements with unattained or unstained resolution. Inspectors also shared the emotional toll of seeing human suffering on a regular basis; the fatalism they felt in trying to help people without support; and the blind eye they learned to give problems that were not specified in the housing code. In describing their work before the program, inspectors stated:

“*So, we see all these social service problems. And there’s really not a lot we can do. It’s terrible… When you’re alone in the house, you see how people live. You see these people living with no food or you see this person that clearly has a mental health issue and, you know, that’s not our job. There’s not a lot we [could] do.*”(Inspector 2).

“*It would put a huge strain on my office. It would put a huge strain on each individual neighborhood. We have these issues that go on forever… And when we’re not working on it… there is no outcome, the neighborhood suffers, the family suffers, everybody suffers.*”(Inspector 3).

#### 3.2.2. Impact of the Program on Inspector’s Work

When asked how the program changed their work, inspectors described increased efficiency and reduced psychological burden as a result of being able to refer residents whose problems would forestall bringing a home to code or whose problems they could not turn from with a clear conscience. They also described providing residents guidance on services when they felt case manager involvement was not required.

“*[The Social Service Referral Program] took away the piece of the inspection for certain types of people that was causing problems for us. If people are going to be evicted or have mental health issues or can’t pay rent–these we can refer to CAPIC… CAPIC has time to case manage people’s needs. We still resolve problems in-house; we just refer our most challenging that are outside the scope of violations.*”(Inspector 1).

“*There’s more free time … so once we do our inspection and write up our report and get CAPIC involved … you don’t have to have it in the back of your head the whole time and check in on it every day… it gives you peace of mind, it releases my inspectors to do other work… It’s a good feeling, takes a lot of stress off us.*”(Inspector 3).

City leadership (the City Manager and Deputy City Manager) also noted the value of being able to rapidly and appropriately define a problem and assemble the needed resources and people from the beginning of an encounter. The program made leadership aware of the psychological toll social problems were exacting on inspectors and the resulting inefficiencies.

“*What’s interesting is that now these problems that were experienced by inspectors, but didn’t necessarily come up to our level, now do… They were simply wrestling with these issues all on their own. Making phone calls, trying to find the resources to address the issues. Sometimes failing, sometimes succeeding, but in a really scattershot way. I think that was sort of an invisible workload down in [The Inspectional Services Department] that has now become part of an understood process of engagement… and given a form and a direction on how to solve these problem situations.*”(City Leader 1).

The total number of referrals in the first eight months of the program was fewer than City leadership anticipated. When asked about the number of referrals, inspectors shared that they only referred the most “hopeless” and time-consuming cases, and that even this small number was valuable to them.

“*When I challenge [inspectors] on the number of cases… it’s the intractable cases that can be time eaters for inspectors and endanger the client… So, I would say it makes [inspectors] more effective and therefore engenders a sense of actually assisting people to a solution that’s helpful to [residents] rather than punitive.*”(City Leader 1).

“*I’m only going to refer people who really need the services–who are destitute or needs can’t be met otherwise.*”(Inspector 1).

#### 3.2.3. Impact on Social Service Provision

For most referrals, the case manager made home visits. In her typical work, clients come to her office. The Social Service Referral Program allowed her to view problems from a new vantage point within homes. She described the value of observing living conditions and building more personal relationships, which contributed to more nuanced assessments of what resources residents had and needed. Often, she found residents needed assistance for several problems, not just the problem that had precipitated a referral. She reported that the residents she helped through the program were unlikely to have connected with CAPIC on their own, especially physically disabled or elderly residents.

“*I spend more time with them [residents referred by inspectors]. Sometimes a client comes in here [to CAPIC] and I’m issuing them a food gift card. I don’t really know about their background… or living situation. I don’t know what their issue is, apart from they’re coming in and they’re telling me, ‘I’m hungry.’ Through [the Social Service Referral Program], it’s a more personal connection I have with the clients.*”(Case Manager).

#### 3.2.4. Impact on Residents and the Community

Inspectors and City leadership described how the program improved residents’ quality of life, reduced the use of costly emergency services, and made neighborhoods safer.

“*When you take the biggest problem of [residents’] lives … and it gives them a better-quality life … dealing with their medical issues, their families, and mental health issues, things like that.*”(Inspector 1).

“*My first thought was to call the police. And now I’m thinking let me get [CAPIC] involved.*”(Inspector 2).

“*If you have one bad building, everything else is so much worse. Once you fix that one building the rest of the neighborhood gets better. Kids are going out and playing on the sidewalk where they weren’t before because the guy had, you know, prostitutes and drug addicts living at his house. So, when he’s kept in check, it changes the whole neighborhood … So, in most of these [cases] we’ve referred … it’s almost like the whole neighborhood had a problem. So, it’s helping the whole neighborhood, not just one person or one person’s life.*”(Inspector 3).

#### 3.2.5. Enabling Environment

Inspectors and City leadership described two key components that enabled the program’s uptake and impact. First, the referral process is easy for inspectors and they see a meaningful difference in their work.

“*As far as I’m concerned, it’s real successful because sometimes it just takes a phone call for me … And it’s like ‘tag, you’re it’ and [the case manager] carries the ball, and she’s really good.*”(Inspector 2).

“*When I raised this as a budgetary issue with [the Director of Inspectional Services] or informally with his staff about moving forward with the program, there is no hesitation on their part … and they wouldn’t simply endorse something because it was there and occasionally it may be okay. So, I trust their judgements when they say to me, ‘This is really important for us to continue’.*”(City Leader 2).

Second, the city contracted an effective and reliable service provider and provided funding to support their work. Inspectors frequently referenced their trust in the timeliness and quality of service CAPIC provides, as well as the benefit of CAPIC’s connections with other service providers in the City. City leadership described the importance of working with the right service provider and that part of what makes the program work is strong personal relationships between the inspectors and the case manager. This relationship did not exist prior to the program but was developed over time as inspectors learned that they could rely on CAPIC and that referring residents decreased their workload.

“*You’ve got to identify the service provider who has the … case management experience to do something like this. … who understands all the connections, that has experience with all the resources that are out there on the state and nonprofit level and is able to bring those in to solve problems. … And on the day-to-day level, certainly making it as simple as possible. [Inspectors] didn’t want to have a complicated reporting system … so shifting the tracking and documentation of case management to the service provider so that it’s not an additional burden on the inspector was important to us.*”(City Leader 1).

#### 3.2.6. Limitations and Generalizability of the Program

The main limitations described by interviewees were that (1) some people that need help refuse to accept help, and (2) some problems could not be addressed either because they were too expensive (e.g., extensive home repairs) or connections to the type of support needed were not readily available through CAPIC’s connections (e.g., legal services). However, to gain buy-in for the program from CAPIC and the inspectors, the requirements for documentation of referrals were limited. City leadership reflected that there was not a process to capture how many residents declined inspectors’ offers of a connection to CAPIC services. While there was a process to capture when residents declined services after meeting with CAPIC, there was not a process to document the number of people that needed a service that could not be provided by CAPIC, nor the type of service that could not be provided.

Despite the limitations, all interviewees felt that the Social Service Referral Program could work in any city. Most added that Chelsea has unique characteristics that they thought contributed to the program’s adoption and success, namely its small geographic size and community-focused culture. No interviewees mentioned the City’s partnership with the Innovation Field Lab when asked what motivated the City to adopt the program. Instead, they described “social ills” that were inadequately addressed through obligation encounters alone and hypothesized that any city with similar problems could benefit.

## 4. Discussion

Integrating a novel Social Service Referral Program within housing inspection allowed for better alignment between the problems residents and inspectors faced and solutions to those problems (Figure 2). When inspectors had a tool beyond citation, they reported that residents received support that not only helped improve housing conditions but also quality of life. When inspectors were able to spend less time on intractable cases, they reported more time for additional housing inspection. When a punitive approach did not fit the problem, inspectors described their pride in helping residents get the care they needed. Though the voice of neighbors and referred residents is missing from this study, inspectors who themselves either live in the community or have spent decades working in the community reported improved quality of life, not only for referred residents, but for entire neighborhoods. Their statements are supported by previous literature on the neighborhood-level impact of helping people meet complex needs and stay housed [38]. As Desmond wrote in the book Evicted, “A single eviction could destabilize multiple city blocks, not only the block from which a family was evicted but also the block to which it begrudgingly relocated [39] (p. 70).” The same may be true of other social crises.

The Social Service Referral Program also modified norms around how case managers at CAPIC engage with clients. Home visits through the program mirror the practice of physician house calls, once the primary mode of healthcare delivery in the U.S. Not only do home visits provide services for a population that might not otherwise access services on their own, but they also give providers an entry point to gather important qualitative information on environmental and social factors that can contribute to or help solve problems. The same is true of physician house calls and other home visiting programs (e.g., Nurse Family Partnership, Early Head Start) [40,41]. Referred residents often received additional services apart from what motivated the initial referral. Half of referred residents were not receiving other forms of assistance at the time of the referral, and for 20%, the referral was their first connection with social assistance. Over half were disabled or elderly. The program delivered services to a segment of the population who was otherwise hard to reach.

Two conditions of an enabling environment for the program were identified through the interviews: (1) A simple referral process that made inspectors’ jobs easier and (2) a trusted, well connected social service agency, funded to carry out its work. The development of a pilot referral program that preceded the formal program also helped facilitate adoption. The program showed that while housing inspection and social service providers employ very different methods, they ultimately have shared goals—housing stability, safety, health, and social welfare. However, these two groups lacked operational alignment. While their integration was facilitated through the Harvard Kennedy School Innovation Field Lab, connections like these do not require a third party to facilitate. However, role reimagining and adaptive leadership are required to achieve alignment between problems and solutions.

### 4.1. Implications

Equipping code enforcement officers, or other frontline staff, to be sensors of at-risk households has the potential to transform the way cities identify and provide services to those in-need. This role re-imagining, within existing structures, can be built upon in other locations and by other professions. By leveraging existing encounters and networks, hard-to-reach or at-risk populations can often be reached more efficiently than through the creation of new channels [42].

Further, we demonstrate how greater value is delivered to the public when enforcement work is problem-oriented to identify and resolve risks through critical engagement, rather than simply applying a rule-driven approach [43]. This type of engagement develops operational capacity to respond holistically and adapt to solve problems. Prior to the program, inspectors’ siloed roles limited their ability to problem-solve. The Social Service Referral program provided inspectors new tools. Seeing the difference that the program made for residents tapped into a sense of purpose beyond code enforcement. Public-service motivation is often lost when city staff lack mechanisms to align solutions with the problems they encounter [16]. By integrating obligation encounters with service provision, we show how housing inspection can renew its connection to public health.

This study represents a steppingstone toward larger, more robust studies on the effectiveness of multisector collaboration to address housing and health problems and other intractable urban health challenges.

### 4.2. Limitations

Inspectors referred only 15 residents to the Social Service Referral Program during the eight-month study period. While inspectors were pleased with this total and City leadership has continued to fund the program, the number of referrals was lower than anticipated at the program’s inception. Prior to the program, inspectors reported encountering at-risk residents on a weekly basis. It is possible that inspectors only referred residents who they thought would accept services (given the high acceptance rate) or when they were confident CAPIC would be able to help—though inspectors denied this type of screening. It is also possible that inspectors felt more equipped and motivated to address less complex social needs on their own through providing information or connections to services without going through the CAPIC case manager, as was described by two inspectors. It is a limitation of the program that there were not specific criterial for what should constitute a referral and a mechanism to document refusals of referrals. It was also not possible to extend the study period past February 2020 due to the COVID-19 pandemic. In March 2020, proactive rental housing inspection ceased. The nature of the referrals and data quality differed substantially after February 2020.

Another important limitation is that it was not possible to interview residents referred to the program due to the COVID-19 pandemic and therefore this evaluation lacks their perspective. Additionally, we cannot quantify the impact of the referral program on the efficiency of housing inspection (e.g., time to resolution of violations; number of inspections per month) because the former is not tracked, and the latter is not available for 2019–2020 due to a change in reporting systems. However, if we use the number of inspections from July 2017 to February 2018 as guidance, approximately 300 housing inspections occurred (excluding tenant turnover inspections when residents are unlikely to be home), only 5% of inspections resulted in a referral.

Among the residents referred, 40% were White (non-Hispanic/Latino). This group makes up only 20% of the total population in Chelsea. The disparity in referrals by race may be due to the age structure of the population. Most elderly residents are White and elderly residents are more likely to be home and encounter housing inspectors [28]. However, it is also possible that inspectors’ decisions to either make or not make referrals reflected implicit and explicit biases or assumptions about eligibility for services.

Lastly, interviewers (first and second authors) were known to the interviewees, having worked together a year prior to the interview. This was both a benefit—because good rapport and candor had been established—but also a limitation. Interviewees may not have felt comfortable critiquing a program the interviewers had helped initiate. To mitigate this, interviewees were encouraged to share their sincere views, both positive and negative and several follow up questions were asked regarding limitations and challenges of the program.

## 5. Conclusions

Fulfilling housing inspector’s mandate to protect the health and safety of residents requires novel approaches. Through the Social Service Referral Program implemented in Chelsea, MA, when inspectors encounter threats to health and safety that cannot be addressed through code enforcement alone, they have a system to link residents to the services they need. This not only allows for timely and appropriate intervention for residents but improves workflow for inspectors. Given the central role that housing plays in the health of individuals and communities, integrating social services within housing code enforcement is a catalyst to improve community health.

## Figures and Tables

**Figure 1 ijerph-18-12014-f001:**
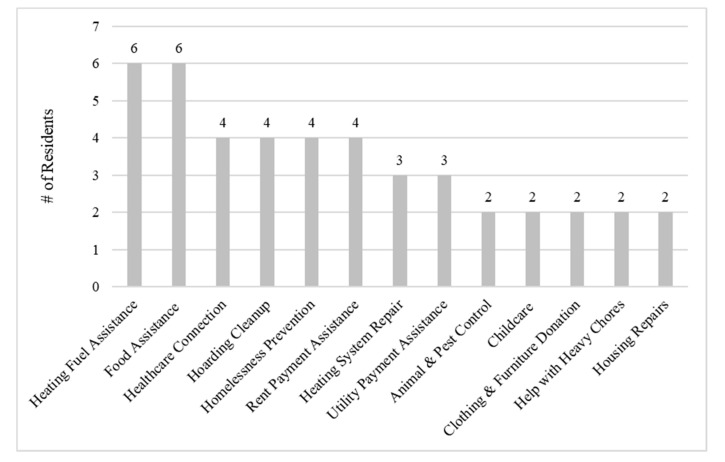
Social services delivered to residents referred by housing inspectors through the Social Service Referral Program.

**Figure 2 ijerph-18-12014-f002:**
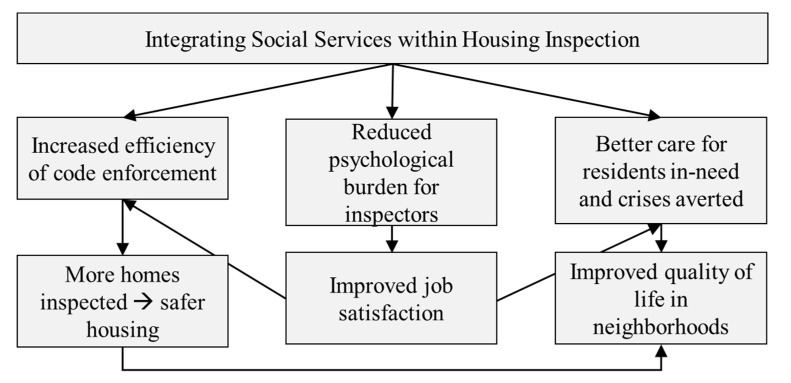
Theory of change for impact of the Social Service Referral Program.

**Table 1 ijerph-18-12014-t001:** Demographics of Residents Referred by Inspectors to Social Services.

Demographic Characteristics	Proportion of Residents Referred to Services ^1^
Sex (female)	53% (8)
Age	
25–39	13% (2)
40–59	20% (3)
60–79	53% (8)
80+	7% (1)
Race	
Asian	7% (1)
Black (not Hispanic/Latino)	7% (1)
Hispanic/Latino	27% (4)
White (Not Hispanic/Latino)	40% (6)
Unknown	20% (3)
Physically disabled	53% (8)
Has children under 5 years	20% (3)
Senior (aged 60+)	60% (9)

^1^*N* = 15.

**Table 2 ijerph-18-12014-t002:** Characteristics of Referrals.

Referral Characteristics	
Referral offer	
Accepted on first offer	60% (9)
Accepted after two or more offers	27% (4)
Declined	7% (1)
Unknown	7% (1)
Connection to social services	
First connection to services	20% (3)
Previously connected to services	33% (5)
Currently receives other services	47% (7)
Number of contacts made between resident and case manager	mean: 9, range: 1–33
Number of service types received	mean: 3, range: 1–5

## Data Availability

The data presented in this study are not publicly available to maintain confidentiality of interviewees and clients of CAPIC. Questions about data access can be directed to the corresponding author.

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
