# Peer review of "Further Inspection: Integrating Housing Code Enforcement and Social Services to Improve Community Health"

_ijerph, 2021, doi:10.3390/ijerph182212014_

Round 1
Reviewer 1 Report
The topic of this manuscript is interesting, but there are also some shortcomings.
First, as an academic paper, the introduction and literature review should be separated into two sections, which is positive to address the literature gap.
Second, more prior studies in the last 3-5 years should be investigated to address the importance of this study.
Third, why only choose the first eight months of the referral program? And the data are only from the first 15 referrals and qualitative interviews with six key informants. Maybe a longer time and more referrals will produce more robust results.
Fourth, the authors are recommended to put forward implications according to the research findings.
Author Response
We are grateful for the astute feedback and the opportunity to improve upon the manuscript. Please see our responses below:
- First, as an academic paper, the introduction and literature review should be separated into two sections, which is positive to address the literature gap.
Thank you for this suggestion. We have added an additional sub-heading to the Introduction section which summarizes our review of the literature on housing and multisector collaboration (section 1.2), followed by our review of the literature placing modern housing inspection in historical context (now section 1.3). We have elected not to include “Literature Review” as it's own a sub-heading or heading.
- Second, more prior studies in the last 3-5 years should be investigated to address the importance of this study.
This is a great point. We have replaced reference 1 on housing as a social determinant of health, dated 1980, with a paper form 2020, S. Rolfe, et al, "Housing as a social determinant of health and wellbeing: Developing an empirically-informed realist theoretical framework," BMC Public Health (2020) doi.org/10.1186/s12889-020-09224-0.
We have also replaced reference 6 on health outcomes associated with poor housing, dated 2011, with two papers from 2020:
Chan, J.H.L., Ma, C.C., 2020. Public Health in the Context of Environment and Housing, in: Fong, B.Y.F., Law, V.T.S., Lee, A. (Eds.), Primary Care Revisited : Interdisciplinary Perspectives for a New Era. Springer, Singapore, pp. 295–310. https://doi.org/10.1007/978-981-15-2521-6_18
Rolfe, S., Garnham, L., Godwin, J., Anderson, I., Seaman, P., Donaldson, C., 2020. Housing as a social determinant of health and wellbeing: developing an empirically-informed realist theoretical framework. BMC Public Health 20, 1138. https://doi.org/10.1186/s12889-020-09224-0
- Third, why only choose the first eight months of the referral program? And the data are only from the first 15 referrals and qualitative interviews with six key informants. Maybe a longer time and more referrals will produce more robust results.
We agree that a longer time period would have allowed for more referral data. We had decided to only include the first 8 months of the program in Summer 2019 due to time constraints. However, those constraints changed after the Covid-19 pandemic began in 2020. Our timeline for interviews was extended to allow time for city staff and social workers to engage in emergency response work. However, we did not extend the timeline for referral data after February 2020 as inspectors stopped routine interior home inspection. From March 2020 through Summer 2021, interior home inspections only took place in the case of emergencies and therefore the referrals generated during this timeframe differ from referrals during times of non-pandemic protocols. Further, the data quality on referrals deteriorated as social workers' time was overwhelmed in responding to other needs during the pandemic.
We have added the following sentence to line 458 in the limitations section:
"It was also not possible to extend the study period past February 2020 due to the COVID-19 pandemic. In March 2020, proactive rental housing inspection ceased. The nature of the referrals and data quality differed substantially after February 2020."
- Fourth, the authors are recommended to put forward implications according to the research findings.
Thank you for this recommendation. We have added a sub-heading in the discussion “4.1 Implications” and modified the paragraph to read,
"Equipping code enforcement officers, or other frontline staff, to be sensors of at-risk households has the potential to transform the way cities identify and provide services to those in-need. This role re-imagining can be built upon in other locations and by other professions. By leveraging existing encounters and networks, hard-to-reach or at-risk populations can often be reached more efficiently than through the creation of new channels [42].
Further, we demonstrate how greater value is delivered to the public when enforcement work is problem-oriented to identify and resolve risks through critical engagement, rather than simply applying a rule-driven approach [43]. This type of engagement develops operational capacity to respond holistically and adapt to solve problems. Prior to the program, inspectors’ siloed roles limited their ability to problem-solve. The Social Service Referral program provided inspectors new tools. Seeing the difference the program made for residents tapped into a sense of purpose beyond code enforcement. Public-service motivation is often lost when city staff lack mechanisms to align solutions with the problems they encounter [16]. By integrating obligation encounters with service provision, we show how housing inspection can renew its connection to public health."
Reviewer 2 Report
This is a very interesting paper looking at the development of a social service referral program using home inspectors as a resource for referrals. I think this is an innovative approach to addressing urban poverty issues. I found the methods employed in the paper appropriate to answer the research question. The conclusions logically flowed from the methods and were appropriately linked back to the literature. Overall, this is a well written, appropriate and very interesting research project. Thank you for the opportunity to review the article.
- The question the research addressed was assessing the effectiveness of a plan that allows housing inspectors to refer at risk individuals to social services that would improve their living conditions. While housing inspectors typically enter a house for housing code enforcement, they are also in a unique situation to identify problems in a household and begin to link at risk households with services that could improve their conditions.
- I found the article to be extremely relevant and interesting. Access to services is one of the major issues facing at risk households and this approach by Chelsea, Massachusetts allows for city officials to identify and refer individuals to services in a nonpunitive manner.
- What does it add to the subject area compared with other published material? The topic is somewhat original. It adds to the literature on social problems of inner cities. One of the major issues in the urban poverty literature is the challenge of providing social services to inner city regions. This article provides an initial assessment of a program that utilizes housing code inspectors as a "sensor" for the city to identify at risk households and begin to provide them access to services that could potentially improve their conditions. I would be interested to see if this approach would work in other settings.
- The paper is well written and easy to read. It does not rely on excessively dense word choices or jargon. I was able to understand the article easily.
- They emphasize the central role of housing at the center of a nexus of health and welfare in the inner city. This puts housing inspectors in the unique position of being able to assess the well being of households. Their conclusions begin to demonstrate the effectiveness of the program in Chelsea.
- I think one of the concerns is with the "completeness" of the research project. The sample size is small and the interviews are one sided, focused on the experience of the housing inspectors exclusively. While I would like to see the experience of the residents included in an assessment, I do not think this should be a reason to reject the research. This study should serve as an initial launch into a potentially powerful and useful research agenda. The ability to use code enforcement officers as a sensor to identify at risk households and provide that information to other city agencies has the potential to transform the way cities identify at risk individuals and help provide them services. I hope others will build on this research, and the other research the authors identified as part of their literature review, to "replicate" this study in other cities and assess its effectiveness. The article submitted here is not the definitive word on the topic, but is one of the initial points of research that will hopefully guide greater research on the topic.
While I can understand criticisms that the research is not robust due to the sample size, I think their approach and initial conclusions are promising and justify publication in an effort to provide an opportunity to replicate the project in other locations. Once this is accomplished, we will be able to build the robust research we all prefer.
Author Response
Thank you very much for your thoughtful review of our study. Despite the small sample size, we hope this innovative approach can serve as a stepping stone for continued work to address urban poverty in other contexts. We are grateful for your comments and perspective!